# Bioinformatics and modelling studies of FhuD, the periplasmic siderophore binding protein from the plant pathogen *Erwinia amylovora*

Sharda Bharti[1☯], Lavinia Carlini[1☯], Alfonso Esposito[2], Stefano Benini [1]*

1 Laboratory of Bioorganic Chemistry and Bio-Crystallography (B₂Cl), Free University of Bozen-Bolzano, Bolzano, Italy, 2 Department of Biotechnology, University of Verona, Verona, Italy

☯ These authors contributed equally to the work.
* stefano.benini@unibz.it

## Abstract

*Erwinia amylovora*, a Gram-negative phytopathogenic bacterium, is the etiological agent of fire blight in apples and pears. Key virulence factors include the exopolysaccharide amylovoran, the type III secretion system, and siderophore-mediated iron uptake. Within the iron uptake pathway, the periplasmic siderophore binding protein FhuD, unique to *Erwinia* species infecting Rosaceae, plays a vital role in transporting iron-loaded siderophores to the inner periplasmic membrane, making it a crucial target for structural and functional characterization. This article presents the predicted 3D model of FhuD from *E. amylovora* (FhuD_Ea), along with the sequence analyses and structural comparison of its homologs from eight organisms whose structures are available in the PDB. We also performed bioinformatics analysis on protein sequences of 145 orthologs. Despite the low sequence identity, the homologs exhibited similar structures, with consistent ligand binding clefts. Nine conserved residues, primarily located in the N-terminal domain, were identified, with the exception of GLY 202 (in the C-terminal domain of FhuD_Ea). Among orthologs, ILE 88 emerged as a notably conserved residue in the N-terminal region, while TRP 64, though often positioned in the binding cleft, was not universally conserved. A phylogenetic tree based on 145 orthologs revealed no distinct grouping between Gram-positive and Gram-negative bacteria, suggesting that the periplasmic binding protein retains similar structural and functional characteristics across diverse bacterial lineages. The apparent lack of universally conserved residues in the ligand-binding pocket suggests functional flexibility, allowing FhuD to recognize siderophores with similar chemical features rather than identical structures. Molecular docking analyses further supported this hypothesis, showing that FhuD_Ea preferentially binds hydroxamate-type siderophores like ferrioxamine, but also accommodates structurally related ligands such as coprogen, with even greater binding affinity. These findings point to an adaptable binding mechanism that may enhance iron acquisition under varying environmental conditions.

**Data availability statement:** Data relevant to this study are available from the Protein Data Bank [https://www.rcsb.org/] (see ID numbers in Table 1 for reference) and UniProt [https://www.uniprot.org/] (ID numbers: P07822, Q47NS2, Q845T3, A0A0H3AJ03, Q76HK0, and Q81L65).

**Funding:** This study was funded by the Libera Università di Bolzano in the form of a Ph.D. fellowship [SB and LC]. The publication of this work was funded by the Open Access Publishing Fund of the Free University of Bozen-Bolzano.

**Competing interests:** The authors have declared that no competing interests exist.

## Introduction

*Erwinia amylovora* is a Gram-negative phytopathogen causing fire blight in rosaceous plants, such as apple and pear [1]. It remains one of the ten most dangerous bacterial plant pathogen due to its economic impact and history of multiple unpredictable outbreaks [2]. *E. amylovora* primarily employs three molecular systems to successfully infect the host plants: (i) secretion of exopolysaccharides (amylovoran and levan); (ii) Type III secretion system (T3SS); and (iii) siderophore-mediated iron uptake [3–5].

Iron is the fourth most abundant element in the earth's crust and exists in two oxidation states, Fe (II) and Fe (III). The redox behavior of iron and its participation in crucial metabolic activities make it valuable to nearly all living organisms except lactic acid bacteria [6]. In aerobic conditions, Fe (II) oxidizes to Fe (III) which forms insoluble ferric hydroxide, thus reducing its bioavailability. To overcome iron starvation, numerous bacteria and fungi secrete low molecular weight iron-chelating compounds called siderophores [7]. Apart from mediating indirect iron uptake in bacteria and fungi, siderophores are also used in different fields including agriculture, ecology, bioremediation, and used as a "Trojan horse" molecule to deliver antibiotics, which might tackle multiple drug resistance [8].

Based on the functional group chelating the iron moiety, siderophores can be broadly classified as hydroxamate, catecholate, carboxylates, polycarboxylates, phenolates, and mixed types [9]. *E. amylovora* secretes only hydroxamate types of siderophores such as desferrioxamine E (DFO-E) along with a small amount of other DFOs (D2, X1-7, and G1-2) [10,11], see Fig 1 for chemical details.

DFO biosynthetic pathway consists of four enzymatic steps involving three proteins DfoJ, DfoA, and DfoC [13,14], which are encoded by a single gene cluster (*dfoJAC* operon) [15]. Siderophores have strong affinity for ferric iron, the Fe(III) form, hence forming a siderophore ferric complex, which enters the Gram-negative bacteria by outer membrane receptors, periplasmic binding proteins (PBP), and inner membrane receptors [16]. In *E. amylovora*, FoxR mediates the entry of iron-loaded siderophore into the periplasm (Fig 2). Once in the periplasm they are guided to an inner periplasmic membrane by ABC transport system (*fhuCDB, hmuSTUV,* and *sitABCD*). All the three ABC transport systems are present in the genome of *E. amylovora* CFBP 1430 [17]. However, unlike *fhuCDB,* the role of *hmuSTUV* and *sitABCD* has not been reported as an desferrioxamine transporter in *E. amylovora.* Within the *fhu* transport system, *fhuD* is present in all the Rosaceae infecting *Erwinia* species studied so far [13] and could be exploited to transport antibiotics into the cells using the Trojan horse approach. These attributes make FhuD from *E. amylovora* (FhuD_Ea) an important target to be thoroughly studied.

We used the predicted 3D structure of FhuD_Ea generated by AlphaFold 3 to compare it with the structure of its homologs from eight different organisms available in the Protein Data Bank (PDB). By superimposition of these homolog structures, we investigated similarities and differences in the binding sites as well as their evolutionary relationship [18]. This study addresses three key aspects of FhuD_Ea: (i)

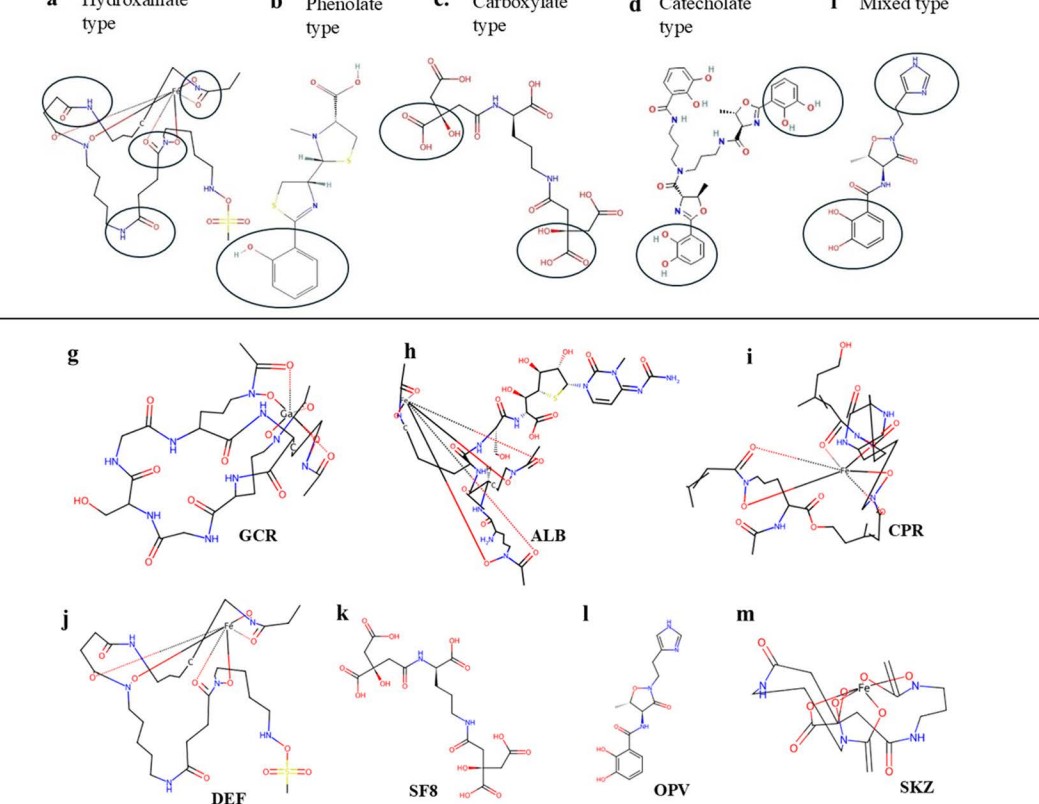

**Fig 1. Representative structures of siderophores categorized by their iron-chelating groups.** (a–f) Different types of siderophores based on their metal-binding functional groups: (a) hydroxamate type, (b) phenolate type, (c) carboxylate type, (d) catecholate type, and (f) mixed-type siderophores. Functional groups are circled. (g–m) Structures of the siderophores included in this study: (g) Gallichrome (GCR), (h) (Albomycin) ALB, (i) Coprogen (CPR), (j) desferal (DEF), (k) SF8, (l) OPV, and (m) (Schizokinen)SKZ, highlighting their coordination with iron ions. The 2D structures of the siderophores were retrieved from PUBChem [12].

comparing its structure with homologs, performing sequence alignment to identify conserved residues in homologs and orthologs, and conducting phylogenetic analysis on Gram-negative and Gram-positive bacteria; (ii) identifying key residues involved in ligand binding in structural homologs deposited in the PDB; and (iii) investigating binding interactions of hydroxamate-type siderophores with FhuD_Ea through molecular docking.

## Methods

### Structure prediction of FhuD_Ea and retrieval of structures of its homologs from PDB

The three-dimensional structure of FhuD_Ea (strain CFBP1430), was retrieved from the AlphaFold 3 server [19,20]. Moreover, the sequences of FhuB (EAMY_2772), FhuC (EAMY_2774), and FhuD (EAMY_2773) were also used to predict the structure of the corresponding protein complex (Fig 3a). The PDB files were visualized in PyMOL [21] to identify the siderophore binding pocket. The FASTA sequence of FhuD served as a query to find the closely related proteins with 3D structures in the PDB using XtalPred [22]. Eighteen protein structures from eight non redundant organisms (*Escherichia coli*, *Staphylococcus aureus*, *Thermobifida fusca*, *Vibrio vulnificus*, *Vibrio cholerae*, *Acinetobacter baumannii*, *Bacillus anthracis*, and *Bacillus cereus*) were identified, and an explanatory table was created to guide a second selection (Table 1).

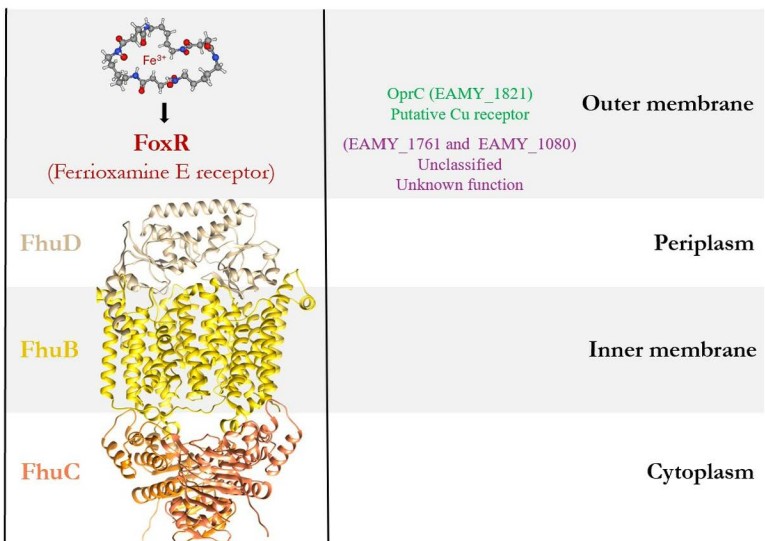

**Fig 2. Schematic representation of siderophore-mediated iron uptake across the bacterial cell envelope.** The diagram depicts the transport pathway of Ferrioxamine E. The outer membrane receptor FoxR specifically recognizes and facilitates the entry of Ferrioxamine E. Also shown are two additional outer membrane receptors: OprC (EAMY_1821), a putative copper receptor, and EAMY_1761 and EAMY_1080, unclassified receptors with unknown function. Following outer membrane translocation, ferrioxamine E is transported through the inner membrane via the FhuCDB, ABC transporter system, comprising FhuD (periplasmic binding protein), FhuB (transmembrane channel), and FhuC (cytoplasmic ATPase).

## Comparison of three-dimensional structures of homologs and analysis of their ligand binding pockets

To compare FhuD_Ea with its homologs we used PDBeFOLD, https://www.ebi.ac.uk/msd-srv/ssm/ [23]. Coordinate file of FhuD_Ea was employed as an input and similar structure were searched using the whole PDB archive with the threshold of lowest acceptance of match of 70%. Precision was kept normal, and the structures were sorted according to the Q value. Out of the eighteen protein structures identified by XtalPred, ten structures from six organisms were selected according to the presence of a ligand; the relevant entries were retrieved from the PDBe database. Ligand annotations from PDBe were used to identify associated ligands, and ChimeraX 1.9 (40) was employed to visualize the spatial arrangement of the binding pockets and the ligand-interacting residues. To highlight evolutionary conservation, residue conservation scores were mapped onto the protein surface using the 'Render by Attribute' tool in Chimera, based on a multiple sequence alignment and visualized on a blue-to-red gradient, with blue indicating low and red indicating high conservation.

## Protein sequence conservation in homologs and orthologs of FhuD_Ea

To identify the most conserved residues in homologous proteins from the eight selected organisms, their FASTA sequences were retrieved from the PDB and aligned using clustalW [31]. The alignment figure was prepared using ESPript 3.0 https://espript.ibcp.fr/ESPript/ESPript/index.php [32]. The amino acids FASTA files for 145 orthologous proteins were retrieved using the orthologs database [33]. Multiple sequence alignments were performed using MUSCLE [34], and a phylogenetic tree was built using the Neighbor-joining method [35]. The tree was tested using 999 replicates of bootstrap [36]. The phylogenetic analyses were performed using the software MEGA (v. 11.0.13) [37] and the tree was plotted using iTOL [38].

## Docking analyses

The interaction between FhuD_Ea and six different compounds, albomycin, coprogen, desferal, desferrioxamine, ferrioxamine, and gallichrome, was explored through molecular docking to enable the prediction of optimal binding

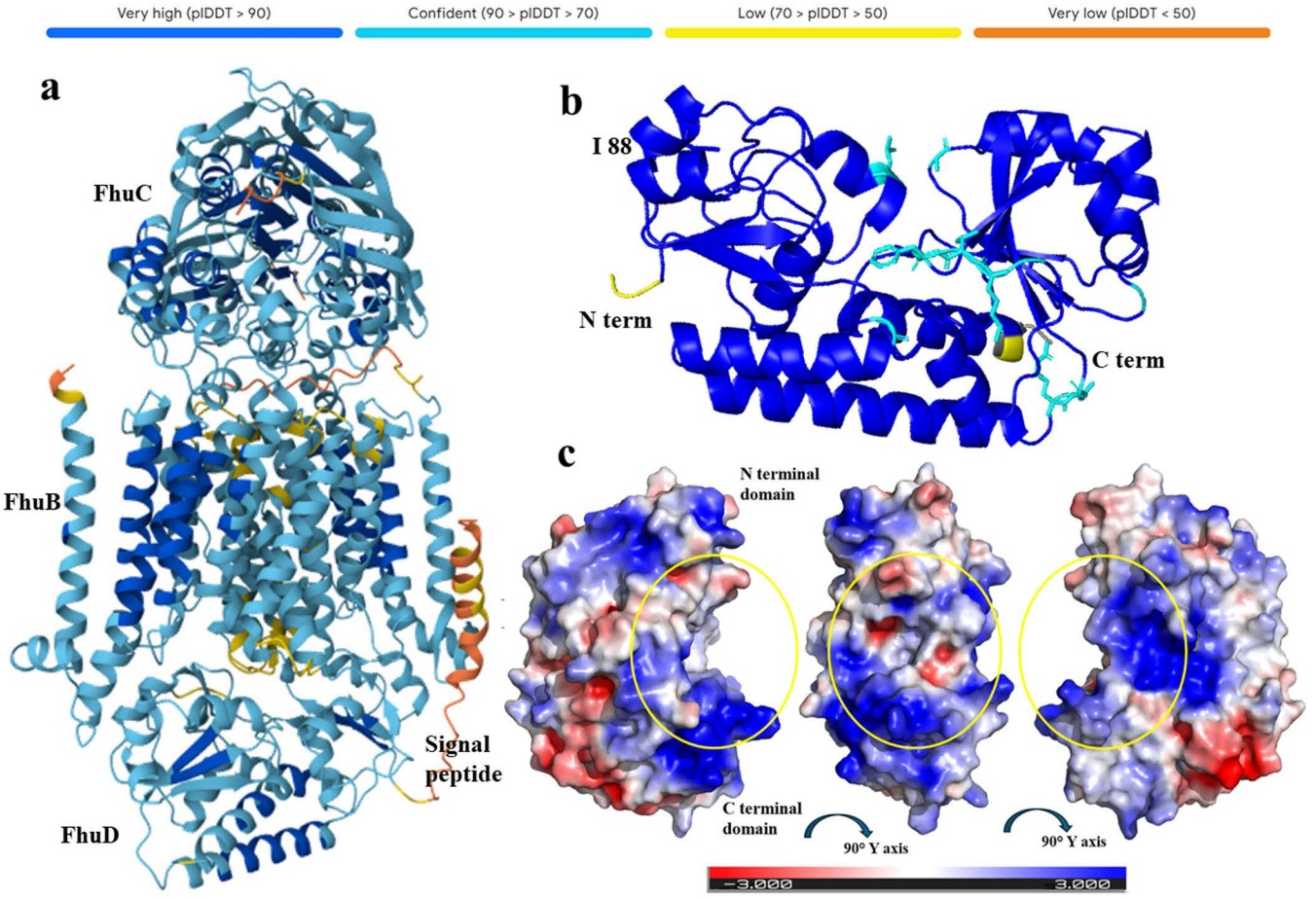

Very high (pIDDT > 90) Confident (90 > pIDDT > 70) Low (70 > pIDDT > 50) Very low (pIDDT < 50)

**Fig 3. Predicted structure of FhuCBD protein complex in *Erwinia amylovora*.** (a) AlphaFold3 model of the FhuCBD complex, showing the spatial organization of the three proteins: FhuC, FhuB, and FhuD. (b) Structure of FhuD, highlighting the N-terminal and C-terminal domains, and the position of residue Ile88. Residues in blue represent regions of very high confidence (pLDDT score 90–100), cyan indicates high confidence (pLDDT 70–90), yellow represents low confidence (pLDDT 50–70), and red indicates very low confidence (pLDDT <50) (c) Electrostatic surface potential map of FhuD_Ea, generated using APBS plugin within PyMOL, depicting positive, negative, and neutral regions in blue, red and white respectively. The circle highlighted in yellow represents the potential ligand binding site.

configurations between a ligand and a macromolecular target [39]. The Autodock Tools v1.5.7 ADT software was utilized to assess potential binding orientations of each compound with the 3D structure of FhuD_Ea. The NCBI PubChem SMILES codes corresponding to each compound were converted into Protein Data Bank (PDB) format using a dedicated converter provided by the NIH (https://cactus.nci.nih.gov/translate/). Subsequently, the PDB files representing the ligands were imported into AutoDockTools v1.5.7 for the docking simulations. To ensure accurate representation, the correct ionization and tautomeric states of amino acid residues of the FhuD_Ea predicted structure were determined by adding hydrogen atoms to the macromolecular structure. Steric clashes were addressed using an appropriate force field, correcting any missing side chains, and minimizing the protein structure. Autogrid4, integrated with AutoDock Tools v1.5.7 ADT, was employed to generate a three-dimensional grid (60 points for each dimension) for mapping the ligands onto the outer and inner surfaces of FhuD_Ea. The grid box was centered on the identified binding pocket with specific adjustment parameters (X = 4.051; Y = −0.980; Z = 4.275, grid centered on the TYR 242), resulting in a resolution of 0.375 Å. For the docking analysis, Autodock4 software utilized the Lamarckian genetic algorithm to facilitate flexible ligand-receptor

**Table 1. List of homologous proteins of FhuD_Ea included in the study.** The table summarizes structurally characterized homologs of FhuD_Ea from various bacterial species. For each entry, the PDB ID, source organism, protein name, co-crystallized ligand (if present), structural determination method, and maximum resolution are reported. Reference numbers correspond to those cited in the main text.

| PDB ID | Organism | Protein | Ligand | Methods | Max. resolution | References |
|---|---|---|---|---|---|---|
| 1EFD | *Escherichia coli* | FhuD | Gallichrome (GCR) | X-ray Cryst. * | 1.9 Å | [24] |
| 1K7S | *E. coli* | FhuD | Albomycin (ALB) | X-ray Cryst. | 2.6 Å | [25] |
| 1ESZ | *E. coli* | FhuD | Coprogen (CPO) | X-ray Cryst. | 2.0 Å | [25] |
| 1K2V | *E. coli* | FhuD | Desferal (DEF) | X-ray Cryst. | 2.0 Å | [25] |
| 7LB8 | *E. coli* | FhuD | – | Cryo-EM** | 3.4 Å | [26] |
| 3EIW | *Staphylococcus aureus* | Native HtsA | Chloride ion | X-ray Cryst. | 1.6 Å | [27] |
| 3LHS | *S. aureus* | HtsA-Fe-SA (open) | Staphyloferrin A (SF8) | X-ray Cryst. | 1.3 Å | [28] |
| 3LI2 | *S. aureus* | HtsA-Fe-SA (closed) | Staphyloferrin A (SF8) | X-ray Cryst. | 2.2 Å | [28] |
| 5DH0 | *Thermobifida fusca* | FscJ_P41 | – | X-ray Cryst. | 2.4 Å | [29] |
| 7W8F | *Vibrio vulnificus* | VatD | Desferal desferrioxamine B (KTY) | X-ray Cryst. | 1.90 Å | To be published |
| 5GGX | *Vibrio cholerae* | FhuD | Deferoxamine mesylate (DEF) | X-ray Cryst. | 3.4 Å | To be published |
| 6MFL | *Acinetobacter baumannii* | BauB | Acinetobactin (OPV) | X-ray Cryst. | 1.9 Å | [30] |
| 6ALL | *Bacillus anthracis* | | | X-ray Cryst. | 2.47 Å | To be published |
| 3TNY | *Bacillus cereus* | YfiY | Schizokinen (SKZ) | X-ray Cryst | 1.55 Å | To be published |

X-ray Cryst.* X-ray Crystallography, Cryo-EM**, Cryogenic Electron Microscopy.

interactions. The results were visualized using the same software, and images were further processed using ChimeraX 1.9 [40]

## Results and discussion

### FhuD_Ea 3D structure and retrieval of its homologs from the PDB

The 3D models of FhuD_Ea and FhuCBD_Ea were generated using the AlphaFold3 server. The FhuD_Ea model exhibits a Predicted Template Modelling (pTM) score of 0.93, indicating a highly reliable prediction, while the FhuCBD_Ea complex has a pTM score of 0.77, suggesting moderate to high confidence in the overall structure. The corresponding Predicted Aligned Error (PAE) maps for both models are provided in the Supplementary Information (S1 Fig) [41]. FhuD_Ea features two domains connected by a hinge region and exhibits significant similarity to known structures of other periplasmic siderophore binding proteins from both Gram-negative and Gram-positive bacteria [24,42]. In FhuD_Ea, most residues have a Predicted Local Distance Difference Test (plDDT) score above 90, except for the 28 amino acids of the signal peptide at the N-terminal and its adjacent residues from ALA 29 to PRO 32. Low confidence regions are also observed at the C-terminal region from ARG 287 to GLU 292. Additionally, six residues (LEU 63, GLU 210, LYS 211, THR 212, PHE 213 and GLY 238) highlighted in cyan in Fig 3b with plDDT values between 70–90. Those six residues are clustered near the potential ligand-binding pocket, located in the cleft between the two domains. Several of these, including THR 212 and PHE 213, are conserved across homologs and potentially involved in ligand recognition through hydrogen bonding and stacking interactions, respectively. Additionally, residues including TRP 64 and TRP 39 and present in the ligand binding pocket of FhuD_Ea are well conserved in the binding pocket of FhuD_Ec. TRP 64 in FhuD_Ea corresponds to TRP 68 in FhuD_Ec, which plays a critical role in binding siderophores such as coprogen, desferal and gallichrome [24]. Similarly, TRP 39 (FhuD_Ea) is spatially aligned with TRP 43 in FhuD_Ec, a key residue in multiple ligand-bound complexes. According to FhuD_Ea electrostatic surface potential map, it is evident that the ligand binding pockets are composed of mostly hydrophobic (white in Fig 3c) and positively charged (blue) residues. Some negatively charged residues are also observed in the deep cleft (red in middle panel of Fig 3c). A similar pattern has been observed in FhuD of *A. baumannii*

binding site [30], where hydrophobic residues predominate in the binding cavity, enabling interaction with chemically diverse xenosiderophores. The varied chemical environment of the binding sites suggests the possibility of accommodating different siderophores, particularly those with hydroxamate groups. Such versatility has been previously reported in periplasmic binding proteins that interact with ligands of varying length and structure (23,24 and [43]. In line with this, *E. amylovora* is known to produce desferrioxamine E ($C_{27}H_{48}N_6O_9$) a hydroxamate-type siderophorecontaining hydroxyl (-OH) and a carbonyl (C=O) groups. These functional groups can make hydrogen bonds with neighboring amino acids, such as ASN 239 in FhuD_Ea, further supporting the protein's capacity to accommodate chemically diverse ligands.

## Comparison of three-dimensional structures of homologs and analysis of their ligand binding pockets

To compare FhuD_Ea with its homologs, a pairwise 3D structural analysis was conducted using PDBeFOLD. FhuD_Ea exhibited the highest resemblance to FhuD_Ec complexed with ligand gallichrome (PDB ID 1EFD), with a RMSD of 1.127, while displaying the lowest similarity to its homolog from *T. fusca*, with an RMSD of 3.094 (5DH0) (Table 2). *E. amylovora* and *E. coli* are both Gram-negative bacteria belonging to Enterobacteriaceae, a possible reason for the highest structural similarity. The overall structural similarity also suggests similar binding pockets for the ligands in both the organisms. Despite the low protein sequence identity, ranging from 53% in *E. coli* to 17% in *T. Fusca* (both versus FhuD_Ea), all the proteins included in this study share similarities at the structural level and the ligand-binding pockets are consistently located within the cleft between the two domains (S2 Fig). Structural similarity of PBP and ligands binding mechanisms has been well described [44–46]. Moreover, the FhuCDB_Ea was compared with FhuBCD_Ec [26]. The overall structures of both complexes are similar, with an RMSD of 5.5. FhuD is positioned in the periplasm, FhuB is located in the inner membrane, and FhuC resides in the cytoplasm facilitating the entry of siderophore iron complex into the cytoplasm. FhuB and FhuD from both organisms exhibit a high degree of structural similarity, while FhuC shows notable variations. Superimposed image of the complexes is shown in S2 Fig.

Despite their overall structural similarity, variations exist in the ligand-binding sites among different organisms. An overview of different ligands binding is presented in Fig 4, with the specific residues involved in binding detailed in Table 3. Residues such as TRP 68, ARG 84, TRP 217, TRP 273, and TYR 275 (based on FhuD_Ec numbering) show strong conservation not only within the Enterobacteriaceae family but also in more distantly related organisms like *S. aureus* and *B. cereus*. These conserved residues are strategically positioned within structurally constrained regions that form the internal surface of the binding pocket. Aromatic and charged residues, such as TRP, TYR, and ARG, are particularly prominent, suggesting a dual role in both maintaining structural integrity and mediating siderophore binding. For instance, *E. coli* and *V. cholerae* share some common residues, such as TRP 68, ARG 84 and TYR 275 (corresponding to TRP 86, ARG 102 and TRP 295 in FhuD of *V. cholerae*), highlighting their likely involvement in π–π stacking interactions with aromatic ligands. In *A. baumannii* residues like

**Table 2. Results of pairwise structural alignments using PDBeFOLD for different homologs of FhuD_Ea.**

| Organisms | PDB ID | Q | RMSD | Nres | Nalign | Ngaps | Seq% |
|---|---|---|---|---|---|---|---|
| *E. coli* | 1EFD | 0.8332 | 1.127 | 261 | 255 | 4 | 53.7 |
| *V. vulnificus* | 7W8F | 0.7363 | 1.370 | 261 | 251 | 7 | 28.3 |
| *V. cholerae* | 5GGX | 0.7281 | 1.503 | 261 | 251 | 5 | 29.1 |
| *E. coli* | 7IB8 | 0.6597 | 1.851 | 261 | 251 | 5 | 52.2 |
| *S. aureus subsp. aureus str. Newman** | 3LHS | 0.4340 | 2.439 | 261 | 234 | 14 | 22.2 |
| *B. cereus** | 3TNY | 0.4277 | 2.576 | 261 | 233 | 14 | 18.0 |
| *A. baumannii* | 6MFL | 0.3993 | 2.561 | 261 | 227 | 13 | 16.7 |
| *T. fusca** | 5DH0 | 0.3657 | 3.094 | 261 | 234 | 15 | 18.8 |

*Gram-positive organisms.

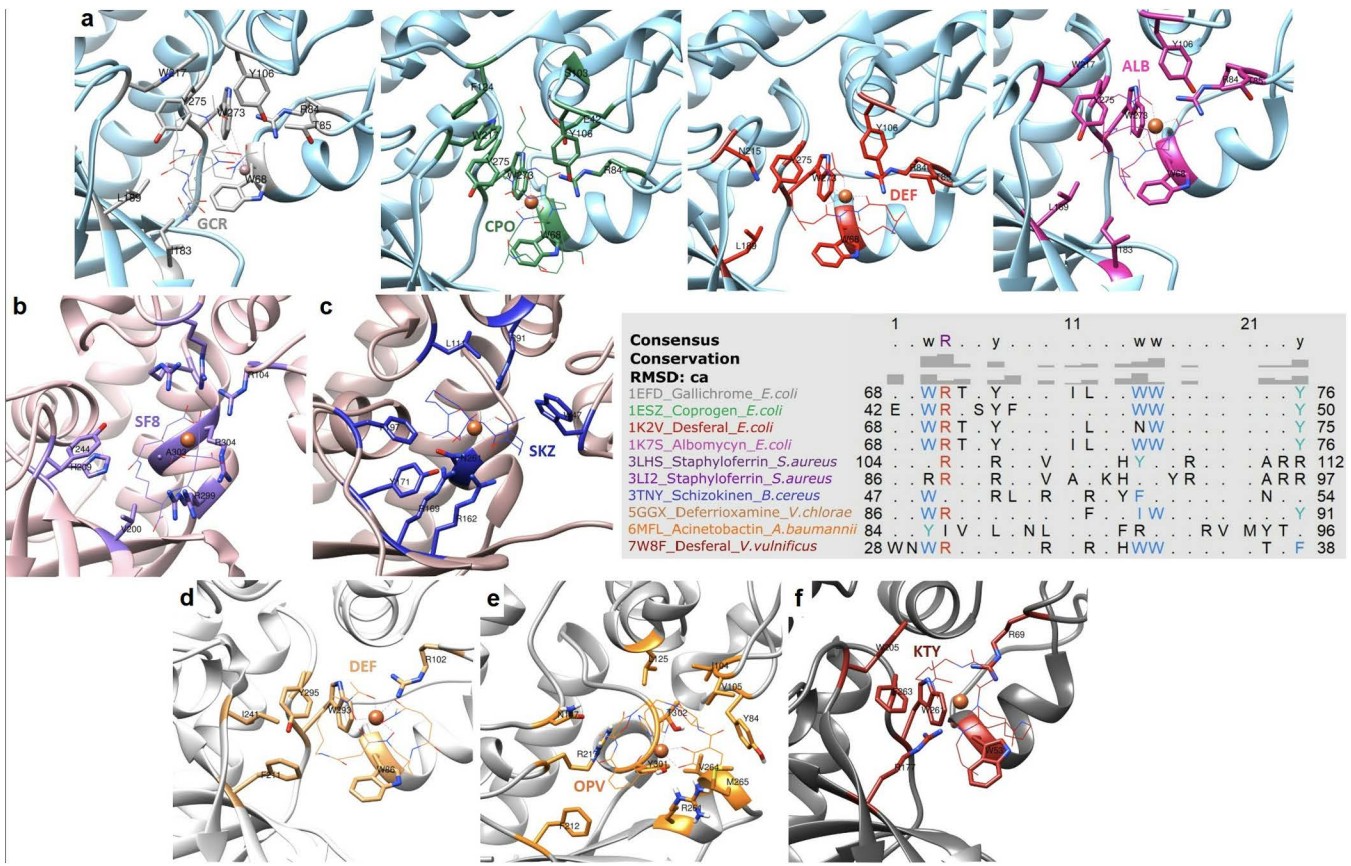

**Fig 4. Overview of ligands binding in FhuD_Ea homologs.** (a) Binding sites of various siderophores in the *E. coli* FhuD structure: Gallichrome (GCR, gray), Coprogen (CPO, green), Desferal (DEF, red), and Albomycin (ALB, pink); (b) Staphyloferrin A (SF8, purple) bound to HtsA from *S. aureus;* (c) Schizokinen (SKZ, blue) binding pocket in YfiY from *B. cereus*; (d) Binding of Desferioxamine (coral) in FhuD from *V. cholerae*; (e) Acinetobactin (OPV, orange) binding site in BauB from *A. baumannii*; (f) Interaction of Desferrioxamine B (KTY, coral) with VatD from *V. vulnificus*.

TRP 293 and ARG 102 are involved across different PDB IDs, while in *S. aureus* subsp. *aureus* N315, specific residues like ARG 86 and ARG 104 are frequently observed. Overall, among all the Gram-negative species included in this study, residues like TRP 43, TRP 68, ARG 84, TYR 106, and PHE 274 are commonly found, whereas in Gram-positive species, ARG 104, ARG 126, PHE 146, and TYR 239 (based on HtsA numbering) were more prevalent. At least one positively charged residue of ARG is present in the binding pocket across all species underscores its functional versatility. Due to the guanidinium group arginine can establish hydrogen bonds as well as cation–π interactions, essential for stabilizing negatively charged siderophore moieties like hydroxamates and catecholates [47]. Moreover, this is supported by prior finding of arginine involvement in binding of siderophore (Staphyloferrin A) by the receptor HtsA in *S. aureus* [28].

Structural data from FhuD_Ec corroborate this conservation: albomycin binding involves residues such as GLU 42, TRP 68, ARG 84, THR 85, SER 103, TYR 106, ILE 183, LEU 189, TRP 217, TRP 273, PHE 274, and TYR 275. In contrast, desferal binds similarly but uniquely engages ASN 64 instead of GLU 42. Notably, ASN 239 is also located in the potential ligand-binding site of FhuD_Ea, with a pLDDT score between 70 and 90. A pLDDT score below 90, suggesting structural flexibility, is typical of ligand-interacting regions. These conserved residues form a functionally significant core within the binding cleft. Aromatic side chains (e.g., TRP 68, TYR 106) likely create a hydrophobic environment favorable for ligand stabilization, while the arginine contributes to a positively charged patch capable of forming electrostatic

**Table 3. Amino acid residues involved in ligands binding in FhuD_Ea homologs in Gram-negative and Gram-positive bacteria, amino acid positions are referred to the position in the sequence of the species.**

| Organisms (Gram-negative) | *E. coli* | | | | *V. cholerae* | *V. vulnificus* | *A. baumannii* |
|---|---|---|---|---|---|---|---|
| PDB ID | 1ESZ | 1EFD | 1K2V | 1K7S | 5GGX | 7W8F | 6MFL |
| Ligands | CPO | GCR | DEF | ALB | DEF | DEF | OPV |
| Residues involved in ligands binding | GLU 42 | **TRP 68** | GLU 42 | TRP 43 | **TRP 86** | TRP 28 | **TYR 84** |
| | TRP 43 | **ARG 84** | TRP 43 | ASN 64 | **ARG 102** | ASN 46 | ILE104 |
| | **TRP 68** | THR 85 | **TRP 68** | TRP 68 | ASP 194 | TRP 53 | VAL 105 |
| | **ARG 84** | TYR 106 | **ARG 84** | ARG 84 | PHE 211 | ARG 69 | LEU 125 |
| | THR 85 | ILE 183 | THR 85 | THR 85 | **TRP 239** | ARG 169 | ASN 147 |
| | SER 103 | LEU 189 | TYR 106 | TYR106 | ILE 241 | ARG 177 | LEU 203 |
| | TYR 106 | **TRP 217** | ILE 183 | ILE183 | **TRP 293** | HIS 179 | PHE 212 |
| | PHE 124 | **TRP 273** | LEU 189 | LEU 189 | TYR 295 | TRP 205 | **ARG 261** |
| | ILE 183 | PHE 274 | ASN 215 | **TRP 217** | | TRP 261 | VAL 264 |
| | LEU 189 | **TYR 275** | **TRP 217** | **TRP 273** | | THR 262 | MET 265 |
| | **TRP 217** | | **TRP 273** | PHE 274 | | PHE 263 | TRP 287 |
| | **TRP 273** | | PHE 274 | **TYR 275** | | | GLY 290 |
| | PHE 274 | | **TYR 275** | | | | TYR 301 |
| | **TYR 275** | | | | | | THR 302 |

| Organisms (Gram-positive) | *S. aureus* subsp. *aureus* str. Newman | | *B. cereus* |
|---|---|---|---|
| PDB ID | 3LHS | 3LI2 | 3TNY |
| Ligands | SF8 | SF8 | SKZ |
| Residues involved in ligands binding | ARG 86 | ARG 86 | TRP 47 |
| | ARG 104 | ARG 104 | ARG 91 |
| | ARG 126 | ARG 126 | LEU 111 |
| | PHE 146 | PHE 146 | ARG 112 |
| | VAL 200 | VAL 200 | **ARG 162** |
| | ALA202 | ALA202 | MET 164 |
| | LYS 203 | LYS 203 | **ARG 169** |
| | HIS 209 | LEU 207 | **TYR 171** |
| | TYR 239 | HIS 209 | **PHE 197** |
| | TYR 244 | TYR 239 | PHE 220 |
| | ARG 299 | TYR 244 | ASN 261 |
| | ALA 303 | ARG 299 | |
| | ARG 304 | ALA 303 | |
| | ARG 306 | ARG 304 | |
| | | ARG 306 | |

interactions and salt bridges. Meanwhile, residues like SER 103 and LEU 189, though less reactive, appear to serve structural roles by maintaining the conformation of key loops and β-strands that define the pocket geometry. Importantly, these residues occupy structurally equivalent positions across homologs when superimposed, indicating not only sequence conservation but also spatial and functional preservation, hallmarks of evolutionary pressure to maintain protein function. This strongly supports the hypothesis that FhuD_Ea shares a conserved mechanism of siderophore recognition with its well-characterized counterparts in *E. coli* and *V. cholerae*, and that disruption of these residues would likely impair binding efficiency and protein stability.

## Sequence conservation in FhuD_Ea homologs and orthologs

The multiple sequence alignment of FhuD_Ea and its homologs from eight different organisms has unveiled 10 highly conserved residues across all the analyzed protein sequences. These residues include LEU 37, GLU 54, ASN 84, GLU 86, PRO 93, LEU 109, ILE 113, ALA 114, PRO 115, and GLY 202. The ten highly conserved residues are highlighted in red boxes in the multiple sequence alignment, while other conserved residues with similar properties are shown in blue boxes (Fig 5). To compare the position of these conserved residues a structural alignment was performed. The conserved residues are highlighted in red and represented as sticks in Fig 6a. Except for GLY 202, all other conserved residues are in the N-terminal domain and are not directly involved in ligand binding. Most of the conserved residues are hydrophobic, likely contributing to proper folding of the protein. In parallel, a comprehensive multiple sequence alignment and structural comparison across diverse bacterial species revealed a second group of conserved residues clustered within the predicted ligand-binding cleft, including TRP 64, ARG 80, TYR 102, SER 178, ILE 186, TRP 203, TRP 214, TRP 270, and TYR 272. Among them, TRP 64 in FhuD_Ea (equivalent to TRP 68 in FhuD_Ec) is notably conserved across orthologs, underscoring its critical role in ligand interaction. To further explore conserved residues, we aligned the protein sequences of 145 orthologs. Results indicate that ILE 88 (FhuD_Ea) is consistently conserved across orthologs (appearing as either LEU, ILE or MET) (S4 Fig). ILE 88 is located in the N-terminal region and does not reside near ligand-binding sites (Fig 6). This suggests a potential role in enhancing protein stability, in agreement with earlier findings that emphasized the importance of conserved isoleucine residues in stabilizing the structure of glutathione transferase [48]. ASN 84, ILE 96 and LEU 224 were also found to be conserved among orthologs. Similar to the homologs, among all the orthologs with the exception of LEU 224, other conserved residues are located in the N-terminal domain. This observation was experimentally supported by the study on the evolution of a modern PBP, where ribose-binding protein (RBP) showed a well-folded N-terminal domain and was expressed in the soluble fraction, while the C-terminal domain was expressed in inclusion bodies due to the presence of disordered regions [46]. Moreover, sequence alignment of RBP with flavodoxin-like proteins showed high sequence similarity in the N-terminal domain [46], suggesting a more conserved and well folded structure of that domain in PBP. This trend was observed thoroughly for both Gram-positive and Gram-negative bacterial species. The phylogenetic tree made with 145 orthologs of FhuD_Ea (S5 Fig), does not show a distinct clustering pattern distinguishing between Gram-positive and Gram-negative species. This result suggests that the protein sequences are highly homoplastic due to rapid evolution or to high rate of horizontal gene transfer.

Although FhuD_Ea is termed as periplasmic protein which facilitates siderophore transport in *E. coli* and other Gram-negative bacteria [24], homologs of FhuD_Ea, albeit under different names, have been discovered in Gram-positive bacteria as well (Table 1). Gram-positive bacteria possess FhuD_Ea homologs, despite their periplasm is much thinner than the one of Gram-negative, such as YfiY (PDB ID; 3TNY) in *B. cereus* and HtsA (3LHS (in open conformation which is similar to apo form), and 3LI2 (closed conformation where the siderophore staphyloferrin A is closely packed between the two domains) in *S. aureus* [28]. The presence of Lipoprotein Permease-like Periplasmic Binding Proteins (Lpps) helps Gram-positive bacteria in transporting substrates across the membrane, typically in association with ATP-binding cassette (ABC) transporters. Many of these Lpps proteins, particularly the substrate-binding proteins (SBPs), carry out functions that are similar to those performed by periplasmic proteins in Gram-negative bacteria [49]. Overall, the finding supports the fact that PBP are ubiquitous among different organisms.

## Docking analyses

Given that periplamic binding proteins (PBPs) are known to accommodate a variety of ligands with different affinities, as exemplified by the putrescine receptor (PotF) from *E. coli*, which binds several biogenic polyamines [45], molecular docking analyses were carried out to investigate the interactions between FhuD_Ea and six siderophores (Fig 7). This included desferrioxamine and ferrioxamine, which are potential ligands for FhuD_Ea, as well as four other hydroxamate-type

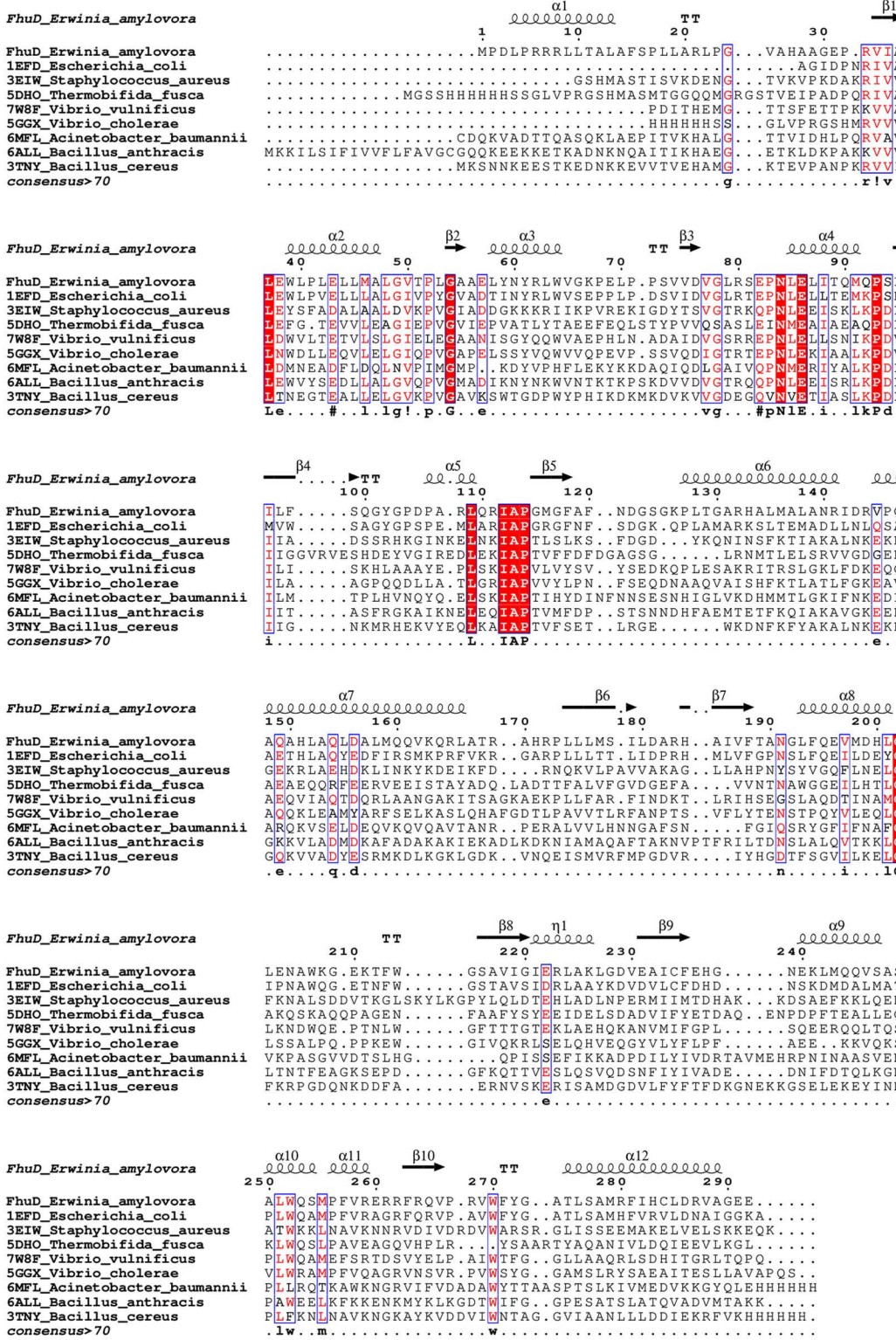

**Fig 5. Multiple sequence alignment of FhuD_Ea and its eight homologs.** Alignment includes protein sequence from *Escherichia coli* (UniProt and PDB ID: P07822, 1EFD)), *Staphylococcus aureus* (SA1979, 3EIW), *Thermobifida fusca* (Q47NS2, 5DH0), *Vibrio vulnificus* (Q845T3, 7W8F), *Vibrio cholerae* (A0A0H3AJ03, 5GGX), *Acinetobacter baumannii* (Q76HK0, 6MFL), *Bacillus anthracis* (Q81L65, 6ALL), and *Bacillus cereus* (3TNY). One sequence

was selected per organism, and the alignment was performed using ClustalW, with images generated using ENDscript3. The secondary structure elements indicated at the top correspond to FhuD_Ea. Strictly conserved residues are highlighted as white characters within red boxes, while similar residues are shown as red characters within white boxes.

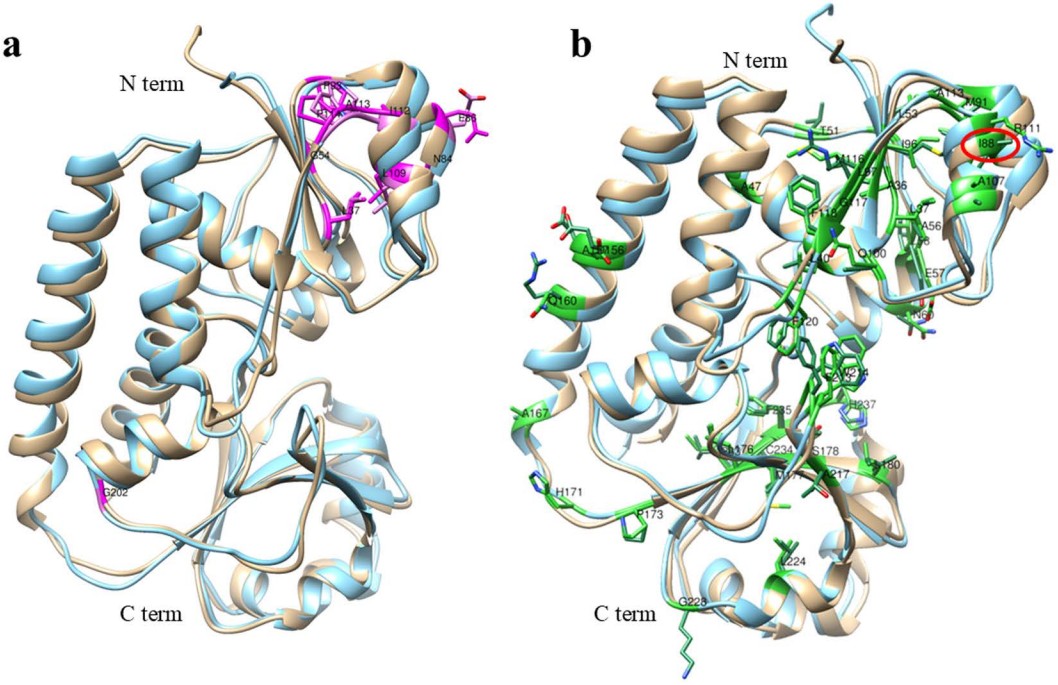

**Fig 6. Superimposed structures of (FhuD_Ea, tan) and its homolog from *E. coli* (PDB ID: 1EFD, cyan).** (a) Ten highly conserved residues across nine homologous sequences are highlighted in magenta, indicating key conserved positions likely important for function or structure. (b) The conserved residues across 145 orthologous sequences are shown in green, emphasizing evolutionary conservation. Among these, ILE88 is specifically marked with a red circle, underscoring its highest conservation.

siderophores (albomycin, coprogen, desferal, and gallichrome) previously shown to interact with FhuD_Ec (Fig 4). The most critical parameter assessed was binding energy, which indicates how strongly each ligand binds to the receptor. Among the tested compounds, coprogen shows the lowest binding energy at −8.16 kcal/mol, followed by ferrioxamine at −7.9 kcal/mol, and desferal at −7.73 kcal/mol, suggesting that coprogen forms the most stable complex with FhuD_Ea. In contrast, gallichrome has the highest binding energy (−5.66 kcal/mol), indicating the weakest interaction with FhuD_Ea.

Ligand efficiency, which measures binding strength normalized to the size of the ligand, further supports these results. Ferrioxamine has the highest efficiency at −0.18 kcal/mol per heavy atom, meaning it binds effectively per unit size. Desferal follows closely at −0.17, while albomycin has the lowest ligand efficiency at −0.10. This implies that despite albomycin forming a moderately stable complex (binding energy of −6.72 kcal/mol), it may not be an optimal binder when considering molecular size.

Inhibition constant (Ki) values, expressed in micromolar (µM) units, confirmed these trends.COP and FO are the most potent inhibitors, with Ki values of 1.04 µM and 1.63 µM, respectively, denoting strong inhibitory potential. Gallichrome on the other hand, has the highest Ki value (71.34 µM), reinforcing its weaker binding affinity compared to the other ligands. Additional docking parameters, such as intermolecular energy, van der Waals, hydrogen bonding, desolvation energy

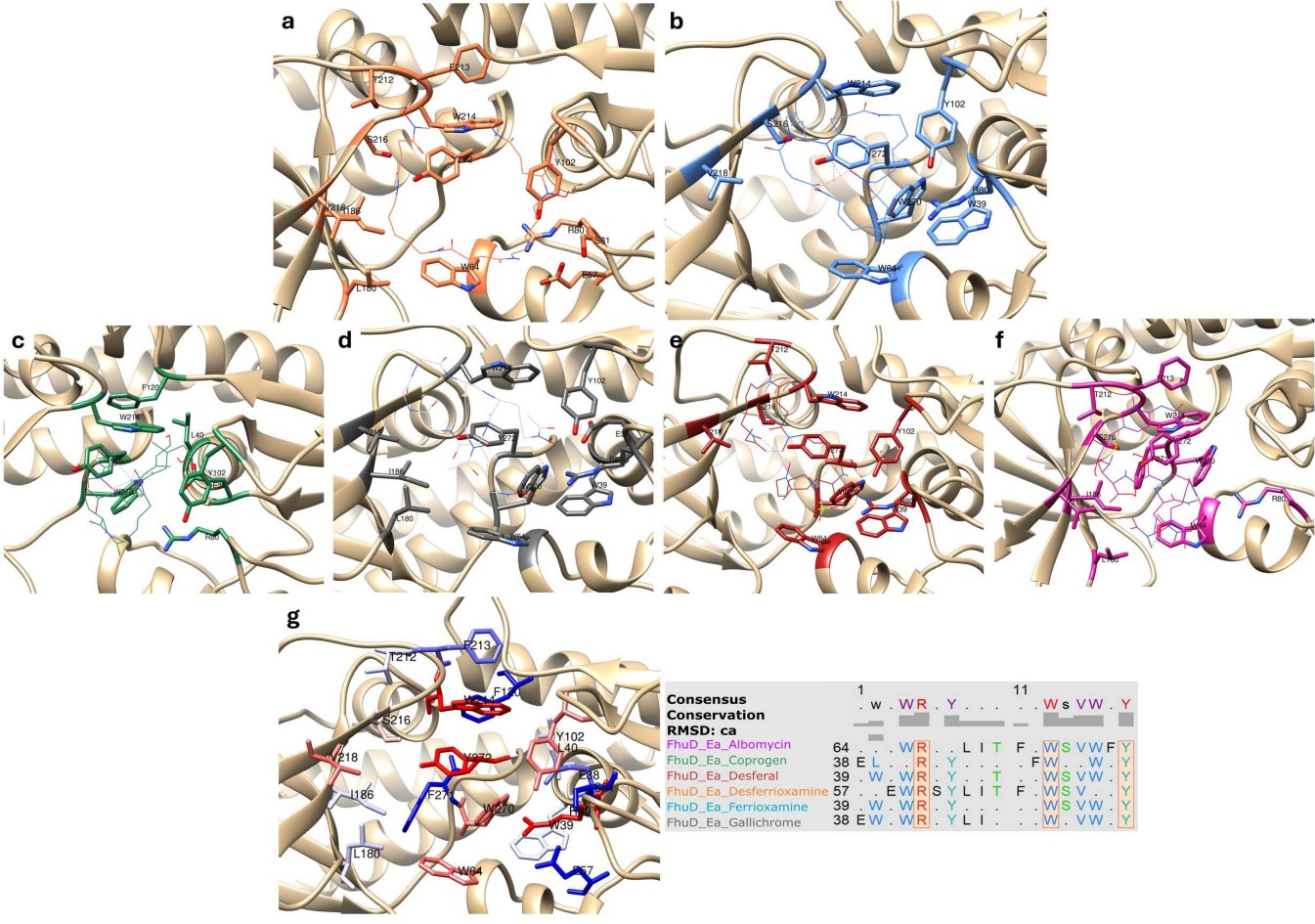

**Fig 7. Docking interactions of FhuD_Ea protein with six different hydroxamate-type siderophores.** Detailed visualizations of ligand–protein interactions for each ligand listed in Table 4. Close-up views of the binding pockets, highlighting key interactions. Interacting residues are clearly labeled with their corresponding residue numbers, and ligands are shown in stick representation for clarity: (a) Desferrioxamine, (b) Ferrioxamine, (c) Coprogen, (d) Gallichrome, (e) Desferal, (f) Albomycin. Panel (g) shows the FhuD_Ea structure color-coded (blue to red) by residue conservation. The accompanying alignment highlights a core set of highly conserved residues shared across FhuD_Ea–ligand complexes, underscoring their potential role in stable siderophore binding.

(vdw_hb_desolv_energy), and electrostatic energy, provide further insights [50]. For instance, coprogen showed the most favorable intermolecular energy (−10.85 kcal/mol), although its relatively high torsional energy (2.68 kcal/mol), indicating some flexibility in its binding conformation, which might influence its binding stability. Overall, coprogen emerges as the most promising binder in terms of stability and inhibitory potential, followed by ferrioxamine and desferal. Gallichrome consistently ranked lowest across most of the evaluated parameters.

Molecular docking results highlight a structurally conserved interaction surface in FhuD_Ea, crucial for siderophore recognition. Sequence alignments across FhuD_Ea–ligand complexes reveal several highly conserved residues – particularly TRP, TYR, and ARG – positioned consistently within the binding pocket (Fig 7g). Their conservation, along with the repeated presence of SER and ARG in ligand-bound conformations, suggests their key role in maintaining structural integrity and mediating specific interactions such as hydrogen bonding and electrostatic contacts. Notably, residues like TRP 64, ARG 80, TRP 214, TYR 272 and SER 216 appear central to stabilizing ligand binding, further supporting the proposed

**Table 4.** Molecular docking parameters for various siderophores (DFO, FO, CPO, GAL, DES, and ALB) with FhuD_Ea.

| Docking Parameters | DFO | FO | CPO | GAL | DES | ALB |
|---|---|---|---|---|---|---|
| Binding energy (kcal/mol) | −5.98 | −7.9 | −8.16 | −5.66 | −7.73 | −6.72 |
| Ligand_efficiency (kcal/mol per heavy atom) | −0.14 | −0.18 | −0.15 | −0.12 | −0.17 | −0.1 |
| Inhib_constant (μM) | 41.18 | 1.63 | 1.04 | 71.34 | 2.17 | 11.85 |
| Intermol_energy (uM) | −6.88 | −7.9 | −10.85 | −9.24 | −10.41 | 11.2 |
| vdw_hb_desolv_energy (kcal/mol) | −6.64 | −7.77 | −10.79 | −9.33 | −10.29 | −11.16 |
| Electrostatic_energy | −0.24 | −0.13 | −0.06 | 0.1 | −0.13 | −0.03 |
| Total_internal(kcal/mol) | −0.1 | 0 | −1.81 | −4.56 | −3.0 | −5.89 |
| Torsional_energy (kcal/mol) | 0.89 | 0 | 2.68 | 3.58 | 2.68 | 4.47 |
| Unbound_energy (kcal/mol) | −0.1 | 0 | −1.81 | −4.56 | 3.0 | −5.89 |
| cRMS (Å) | 0.04 | 0.04 | 0 | 0 | 0 | 0 |
| refRMS (Å) | 9.26 | 9.32 | 9.37 | 29.11 | 10.83 | 9.88 |

binding mechanism. In silico alanine mutagenesis of these key residues resulted in a noticeable reduction in binding affinity and a shift of the ligand toward deeper, less conserved regions of the binding cleft. This displacement emphasizes the importance of both the chemical nature and spatial orientation of the conserved residues in preserving the structural and functional integrity of the FhuD_Ea binding pocket.

*E. amylovora* encodes an outer membrane receptor, FoxR, dedicated to the uptake of ferrioxamine [11,51]. Interestingly, the periplasmic binding protein (FhuD_Ea) appears capable of binding multiple structurally related siderophores, indicating that the specificity for siderophore uptake is primarily regulated at the outer membrane level. In contrast, the periplasmic proteins may adapt to different ligands by modifying a few residues within their binding pockets.

The *E. amylovora* genome includes four TonB-dependent receptors: OprC (EAMY_1821), a putative copper receptor; FoxR, the ferrioxamine receptor; and two unclassified receptors (EAMY_1761 and EAMY_1080) [52]. If one of these unclassified receptors facilitates coprogen uptake, the FhuD_Ea would likely bind it effectively, as supported by our docking studies, thereby offering an alternative iron uptake mechanism. This adaptive capacity would be particularly advantageous in scenarios of cross-contamination with competing microbes, such as *Alternaria alternata*, a pathogen that causes leaf blotch and fruit spot in apples [53] and produces coprogen [54].

## Conclusion

This study focuses on understanding the structure and ligand-binding properties of the periplasmic binding protein FhuD in *Erwinia amylovora*, a phytopathogen causing fire blight in apple and pear. By integrating bioinformatics tools, structural modeling, and molecular docking, we reconstructed the 3D structure of FhuD_Ea and compared it with homologs and orthologs from diverse bacterial species. Despite low sequence identity among homologs, structural conservation was evident, particularly in the overall domain architecture and the position of the ligand-binding cleft. The binding site of FhuD_Ea comprises mainly hydrophobic residues alongside several strategically placed positively charged residues, such as arginine, which are highly conserved among both Gram-positive and Gram-negative homologs. Molecular docking studies suggest that FhuD_Ea exhibits a clear preferential affinity for hydroxamate-type siderophores like ferrioxamine, but it also demonstrates flexibility in binding other structurally related compounds. Notably, coprogen demonstrated the strongest binding affinity, implying that FhuD_Ea may adapt its binding to optimize iron uptake under competitive or variable environmental conditions. Sequence alignments and docking-based residue mapping revealed a structurally conserved interaction surface, with residues such as TRP 64, ARG 80, TYR 102, SER 178, ILE 186, TRP 203 TRP 214, TRP 270 and TYR 272 playing central roles in stabilizing ligand interactions. These findings support the hypothesis that periplasmic

binding proteins like FhuD_Ea rely on a conserved structural framework to accommodate chemically similar siderophores, even in the absence of strict sequence conservation.

This work advances our understanding of siderophore recognition in *E. amylovora*, shedding light on the adaptability of its iron acquisition system. However, while these findings offer valuable insights into potential binding interactions and structural features, they remain hypothetical. Experimental validation through recombinant expression of FhuD_Ea followed by binding assays is necessary to confirm these interactions. Future experimental efforts, particularly those involving site-directed mutagenesis of conserved residues, such as ILE 88 in the N-terminal domain or the structurally critical residues within the binding cleft, will be crucial for clarifying their roles in protein folding and their indirect influence on ligand binding dynamics.

## Supporting information

**S1 Fig. Predicted Aligned Error map of FhuCDB_Ea and FhuD_Ea.**
(PDF)

**S2 Fig. Superimposed FhuD_Ea homologs included in this study highlighting the ligand binding pocket and superimposed complex FhuCDB from *E. coli* and *E. amylovora*.**
(PDF)

**S3 Fig. Detailed visualization of ligand–residue interactions.**
(PDF)

**S4 Fig. Multiple sequence alignment of FhuD_Ea orthologs.**
(PDF)

**S5 Fig. Phylogenetic tree of FhuD_Ea orthologs.**
(PDF)

## Author contributions

**Conceptualization:** Stefano Benini.

**Data curation:** Sharda Bharti, Lavinia Carlini.

**Formal analysis:** Sharda Bharti, Lavinia Carlini, Alfonso Esposito.

**Investigation:** Sharda Bharti, Lavinia Carlini, Alfonso Esposito.

**Project administration:** Stefano Benini.

**Supervision:** Stefano Benini.

**Writing – original draft:** Sharda Bharti.

**Writing – review & editing:** Sharda Bharti, Lavinia Carlini, Alfonso Esposito, Stefano Benini.

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
