## [Decision Letter · Decision Letter 0]

PONE-D-25-04770Bioinformatics and modelling studies of FhuD, the periplasmic siderophore binding protein from the plant pathogen Erwinia amylovoraPLOS ONE

Dear Dr. Benini,

Thank you for submitting your manuscript to PLOS ONE. After careful consideration, we feel that it has merit but does not fully meet PLOS ONE’s publication criteria as it currently stands. Therefore, we invite you to submit a revised version of the manuscript that addresses the points raised during the review process.

We look forward to receiving your revised manuscript.

Kind regards,

Abhijeet Shankar Kashyap

Academic Editor

PLOS ONE

Journal Requirements:

2.  Please update your submission to use the PLOS LaTeX template. The template and more information on our requirements for LaTeX submissions can be found at http://journals.plos.org/plosone/s/latex .

Reviewers' comments:

Reviewer's Responses to Questions

**Comments to the Author**

1. Is the manuscript technically sound, and do the data support the conclusions?

Reviewer #1: Partly

Reviewer #2: Yes

Reviewer #3: Partly

2. Has the statistical analysis been performed appropriately and rigorously? 

Reviewer #1: N/A

Reviewer #2: N/A

Reviewer #3: Yes

3. Have the authors made all data underlying the findings in their manuscript fully available?

Reviewer #1: Yes

Reviewer #2: Yes

Reviewer #3: Yes

4. Is the manuscript presented in an intelligible fashion and written in standard English?

Reviewer #1: Yes

Reviewer #2: Yes

Reviewer #3: Yes

5. Review Comments to the Author

Reviewer #1: In this manuscript entitled “Bioinformatics and modelling studies of FhuD, the periplasmic siderophore binding protein from the plant pathogen Erwinia amylovora”, the team led by Stefano Benini used AlphaFold to model the FhuD protein and performed sequence analyses and structural comparison (including molecular docking) against its homologs from eight organisms with available PDB entries, along with sequence alignment and phylogenetic analysis of 145 orthologs. Their findings revealed some structural features of siderophore binding in gram-negative and gram-positive bacteria, i.e., a similar binding pocket for various ligands that is constructed by variable key residues. The information is useful and will contribute to our understanding about the structural basis of ligand binding.

To improve on the clarity, please address the following concerns/suggestions:

Ile 88 was singled out as a key residue (Abstract and Discussion), but the supporting evidence and argument is missing. The authors need to present their logic clearly and potentially label Ile 88 in all relevant figures (Fig 1a-c, 2a-d, 4a-c, 5a-f) to support their argument.

Please correct the disagreements between text and figures listed below:

Page 9, line 143: “19 residues highlighted in cyan” is not shown in Fig 1a.

Page 10, line 156: “Fig 1d” is missing (not submitted for figure)

Page 12-14: All discussion and legend related to “Figure 2” doesn’t include 2e and 2f as shown. Were the latter two necessary for this study?

Page 16, line 230-231: “The conserved residues are highlighted in red .. in Fig 5A” is not the case.

Please correct other issues listed below:

Page 9, line 149-150: rephrase the sentence to avoid potential confusion

Page 16, line 226: “Glu 90” should be “Glu 86”

Page 16, line 231-232: Figure 5A is mentioned before Figure 4A. It is better to reverse the order.

Page 16, line 236: take out “Ile 88” from “Other residues …”

Page 19, Table 3: remove extra # on top right

Reviewer #2: The manuscript “Bioinformatics and modelling studies of FhuD, the periplasmic siderophore binding protein from the plant pathogen Erwinia amylovora” is dedicated to the sequence and structural comparison of the high score AF FhuD model with its structural and sequence homologs. By sequence alignments and residue conservation analysis, the authors revealed FhuD siderophore binding site, which was followed by the ligand docking targeted to the site. Knowledge of structur and functional details for EaFlXD, which plays a vital role in transporting iron-loaded siderophores to the inner periplasmic membrane

so as details on its ligands specificity may help to develop the treatment against Erwinia amylovora, causing fire blight disease affecting apples and pears.

Below are suggestions which benefit the paper.

Use the same AlphaFold version everywhere in the text when referring to how your model was generated for consistency.

It would be beneficial to have a figure with chemical details of possible classes of siderophores, as mentioned in the introduction to the general reader (page 3, line 53).

Removing redundant entries from the tables 1 and 2 will benefit both tables.

The binding partner for FhuB interacting with FhuD is missing from the table 1.

Removal of non-relevant to siderophore binding site ligands like ACT (which is acetate, not acetone as mentioned in the figure 2 caption), EDO (ethylene glycol) or Cl from the figure 2, table 2,3 and discussion will benefit the paper.

The text is not easy to follow due to many unimportant details. In the chapter dedicated to the comparison of three-dimensional structures of homologs and analysis of their ligand binding pockets, it will be beneficial to follow the same structure description used for E.coli homolog (N-lobe, C-lobe and the linker ).

Include the references to the subfigures of Figure 2a,b,c, etc, in the text.

Figure 2 doesn’t have enough details on the ligands and ligand binding residues and their interactions; residue numbers are missing, as are the ligand residue interactions. Residue labels with numbers are also needed. The subpanels of the figures could follow the same style centered around the ligand binding pocket. The figure caption is far from the journal quality.

Table 3 doesn’t have the bonding details either, which makes you wonder about the roles of the multiple residues mentioned in the table.

Figure 3. The figure caption for needs uniport IDs, not PDB IDs, it is showing sequence alignments.

Figure 4. You already introduced the siderophore binding site in Figure 2; you can color the molecule by residue conservation and show it in 2 different orientations (chimera has this option).

Table 3, please place the units associated with each row for binding energy, ligand efficiency etc.

Create a supplementary figure with details of the ligand protein interactions for each ligand in table 3.

EaFluD has 53% similarity with E.coli FhuD, which was recently characterised as part of a ferrichrome importer FhuCDB from E. coli (https://doi.org/10.1038/s42003-021-02916-2). Aplhafold 4 is reasonable with protein complexes. Could you generate the Erwinia amylovora FhuCDB model? Will it be any differences with E.coli one?

Reviewer #3: Overall Summary:

Bharti et al. present a manuscript that evaluates predicted protein structure for an E. amylovora siderophore binding protein FhuD and binding analysis to a set of known siderophores. The manuscript uses alpha fold for structural predictions and then uses docking through autodock to compare FhuD to 6 siderophores. While the manuscript does a thorough analysis of the predicted structure of FhuD and the phylogenetic differences across various homologs, the docking predictions to the set of siderophores is not validated by binding analysis in vitro. Further, there is also a lack of the evaluation of a larger set of proteins from the E. amylovora genome for binding potential to the set of siderophores tested. These sets of information will be necessary to validate the docking predictions. Comments are specifically provided below.

•Table 3/Figure5: While these are predicted binding for the different ligands to FhuD, it needs to be validated by the recombinant expression of the FhuD and binding assays. Currently these are just predictions.

•Figure 2/ Figure 5: Since the manuscript does not include data about the validated binding of FhuD to these ligands, another exercise would be to selectively modify the protein sequence to alter the potential binding pockets and rerun this through alpha fold and Autodock. To check if this can alter the binding as per the docking analysis.

•Page 15 and Page 19 both have a Table 3. This might be a typo that needs to be modified.

•Figure 4: While FhuD is the main transporter of interest in this study, due to the restricted nature of docking to a set group of siderophores, the data is mainly predictive and needs to be validated by binding studies. However, this data could be also supplemented by reverse docking analysis where the alpha fold predictions of the E. amylovora CFBP1430 genomic sequence can then be the protein database that each of the ligands get reverse docked into. This helps provide an unbiased outlook of the best candidates out of the E. amylovora genome that can bind to these siderophores.

•Table 4: due to just the evaluation of FhuD docked against 6 targets, the binding energy and efficiency is present without any context. These either need to be supplemented with binding assays or reverse docking, otherwise it is unclear how this interaction potential will vary if a larger group of proteins were evaluated.

6. PLOS authors have the option to publish the peer review history of their article (what does this mean? ). If published, this will include your full peer review and any attached files.

**Do you want your identity to be public for this peer review?** For information about this choice, including consent withdrawal, please see our Privacy Policy .

Reviewer #1: No

Reviewer #2: No

Reviewer #3: No

---

## [Author Response · Author response to Decision Letter 1]

14 May 2025

We thank both Reviewers for providing useful suggestions and comments which helped us improve the manuscript. We believe to have answered all the issues raised by the Reviewers by making the appropriate changes and implementations. We hope that our manuscript has now reached the level of quality necessary to be accepted for publication.

Answers to the questions/comments are in boldface

Review Comments to the Author:

Reviewer #1: In this manuscript entitled “Bioinformatics and modelling studies of FhuD, the periplasmic siderophore binding protein from the plant pathogen Erwinia amylovora”, the team led by Stefano Benini used AlphaFold to model the FhuD protein and performed sequence analyses and structural comparison (including molecular docking) against its homologs from eight organisms with available PDB entries, along with sequence alignment and phylogenetic analysis of 145 orthologs. Their findings revealed some structural features of siderophore binding in gram-negative and gram-positive bacteria, i.e., a similar binding pocket for various ligands that is constructed by variable key residues. The information is useful and will contribute to our understanding about the structural basis of ligand binding.

To improve on the clarity, please address the following concerns/suggestions:

Ile 88 was singled out as a key residue (Abstract and Discussion), but the supporting evidence and argument is missing. The authors need to present their logic clearly and potentially label Ile 88 in all relevant figures (Fig 1a-c, 2a-d, 4a-c, 5a-f) to support their argument.

R: ILE has been highlighted in Figures 3b and 6b. Additionally, relevant literature mentioning the structural importance of conserved isoleucine residues has been cited and included in the discussion to strengthen the result. (Note: Two figures have been added in the revised manuscript, so the original Figure 1 has been renumbered as Figure 3)

Please correct the disagreements between text and figures listed below:

- Page 9, line 143: “19 residues highlighted in cyan” is not shown in Fig 1a.

R: The structure has been revised using AlphaFold3, replacing the earlier AlphaFold2 model. As a result, 13 residues (instead of 19 previously) now exhibit pLDDT scores between 70 and 90. These residues are highlighted in cyan and displayed as sticks to enhance their visibility. (Note: Figure 1a in the initial draft corresponds to Figure 3b in the revised manuscript.)

- Page 10, line 156: “Fig 1d” is missing (not submitted for figure)

R: The disagreement has been revised and corrected.

- Page 12-14: All discussion and legend related to “Figure 2” doesn’t include 2e and 2f as shown. Were the latter two necessary for this study?

R: The image has been modified to make it clearer: each structure has been placed in a different panel, the residues involved in binding are now labelled and colored differently for each ligand. Panel d has been removed, replacing it only with the more representative e. The caption has also been changed to better clarify what is in each panel.

- Page 16, line 230-231: “The conserved residues are highlighted in red .. in Fig 5A” is not the case.

R: The figure we were referring to was 4, the error has been corrected.

- Page 9, line 149-150: rephrase the sentence to avoid potential confusion

R: Text has been paraphrased.

Please correct other issues listed below:

- Page 16, line 226: “Glu 90” should be “Glu 86”

- Page 16, line 231-232: Figure 5A is mentioned before Figure 4A. It is better to reverse the order.

- Page 16, line 236: take out “Ile 88” from “Other residues …”

- Page 19, Table 3: remove extra # on top right

R: The issues listed were promptly addressed and corrections were ensured.

Reviewer #2: The manuscript “Bioinformatics and modelling studies of FhuD, the periplasmic siderophore binding protein from the plant pathogen Erwinia amylovora” is dedicated to the sequence and structural comparison of the high score AF FhuD model with its structural and sequence homologs. By sequence alignments and residue conservation analysis, the authors revealed FhuD siderophore binding site, which was followed by the ligand docking targeted to the site. Knowledge of structur and functional details for EaFlXD, which plays a vital role in transporting iron-loaded siderophores to the inner periplasmic membrane

so as details on its ligands specificity may help to develop the treatment against Erwinia amylovora, causing fire blight disease affecting apples and pears.

Below are suggestions which benefit the paper.

Use the same AlphaFold version everywhere in the text when referring to how your model was generated for consistency.

R: All the structures have now been generated using AlphaFold3, and the corresponding figures and related descriptions have been updated accordingly to reflect this change.

It would be beneficial to have a figure with chemical details of possible classes of siderophores, as mentioned in the introduction to the general reader (page 3, line 53).

R: A figure depicting the major classes of siderophores, with key functional groups highlighted, has been added. Additionally, the structures of the ligands used in this study have been included.

Removing redundant entries from the tables 1 and 2 will benefit both tables.

R: Redundant entries were removed from Tables 2 and 3, as they were not included in the analyses. However, they were retained in Table 1 to document all entries initially identified.

The binding partner for FhuB interacting with FhuD is missing from the table 1.

R: We limited the table to proteins structurally similar to FhuD from E. amylovora to maintain clarity. The interaction and the predicted formation of a potential complex are discussed in the Introduction.

Removal of non-relevant to siderophore binding site ligands like ACT (which is acetate, not acetone as mentioned in the figure 2 caption), EDO (ethylene glycol) or Cl from the figure 2, table 2,3 and discussion will benefit the paper.

R: Redundant entries not bound to a siderophore have been removed, and only the non-redundant ones, as well as those bound to a siderophore, have been retained. (The mention of 'acetone' was likely due to an autocorrect error.)

The text is not easy to follow due to many unimportant details. In the chapter dedicated to the comparison of three-dimensional structures of homologs and analysis of their ligand binding pockets, it will be beneficial to follow the same structure description used for E.coli homolog (N-lobe, C-lobe and the linker).

R: The text has been revised to improve clarity and readability, with a focus on making the key findings more accessible.

Include the references to the subfigures of Figure 2a,b,c, etc, in the text.

R: The subsection of Figure 4 highlighting the ligand-binding residues has been referenced and emphasized in the main text accordingly (Note: Figure 2 has been renumbered as Figure 4)

Figure 2 doesn’t have enough details on the ligands and ligand binding residues and their interactions; residue numbers are missing, as are the ligand residue interactions. Residue labels with numbers are also needed. The subpanels of the figures could follow the same style centered around the ligand binding pocket. The figure caption is far from the journal quality.

R: The figure has been thoroughly revised to address the lack of detail regarding ligands, binding residues, and their interactions. Residue numbers and specific ligand-residue contacts have now been included and clearly labeled. Each structure is displayed in a separate panel, with consistent formatting focused on the ligand binding pocket, and residues are color-coded and annotated accordingly. Moreover, a supplementary file has been added to illustrate the interactions between residues and ligand molecules

Table 3 doesn’t have the bonding details either, which makes you wonder about the roles of the multiple residues mentioned in the table.

R: The role of the residues listed in Table 3 has been clarified in the main text, where we specify their involvement in the interaction surface of FhuD_Ea based on conservation analysis and structural alignment with homologs. These residues are discussed in detail with reference to their predicted positions and potential contributions to ligand recognition.

Figure 3. The figure caption for needs Uniprot IDs, not PDB IDs, it is showing sequence alignments.

R: UniProt IDs have been included alongside PDB IDs for all sequences.

Figure 4. You already introduced the siderophore binding site in Figure 2; you can color the molecule by residue conservation and show it in 2 different orientations (chimera has this option).

R: Residue conservation scores were mapped onto the protein surface using the "Render by Attribute" tool in Chimera, based on a multiple sequence alignment.

These alignments highlight a set of highly conserved residues across both FhuD homologs and FhuD_Ea–ligand complexes, particularly tryptophan, tyrosine, and arginine, which are consistently preserved at structurally equivalent positions. Their conservation across diverse species and ligand contexts suggests a fundamental role in maintaining the structural integrity of the binding pocket and in mediating stable interactions with siderophores. Additionally, the recurrence of serine and arginine in key positions, especially in ligand-bound forms, indicates they may contribute to ligand specificity and recognition through hydrogen bonding or electrostatic interactions. These conserved features reinforce the hypothesis that FhuD_Ea engages siderophores through a structurally conserved and functionally relevant interaction surface.

Table 3, please place the units associated with each row for binding energy, ligand efficiency etc.

R: The table has been updated to include units for all relevant values. Binding energy is now reported in kcal/mol, ligand efficiency in kcal/mol, and all other parameters have their corresponding units clearly indicated to ensure clarity and consistency.

Create a supplementary figure with details of the ligand protein interactions for each ligand in table 4.

R: As requested, a supplementary figure has been created to provide detailed visualizations of the ligand–protein interactions for each ligand listed in Table 4. This figure includes close-up views of the binding sites, with annotated hydrogen bonds, hydrophobic contacts, and interacting residues labeled with their corresponding numbers.

EaFhuD has 53% similarity with E.coli FhuD, which was recently characterised as part of a ferrichrome importer FhuCDB from E. coli (https://doi.org/10.1038/s42003-021-02916-2). Aplhafold 4 is reasonable with protein complexes. Could you generate the Erwinia amylovora FhuCDB model? Will it be any differences with E.coli one?

R: fhuB (EAMY_2772), fhuC (EAMY_2774), fhuD (EAMY_2773) sequences were used for the prediction of the structure of the complex, which was depicted in Figure 3a. Additionally, the FhuBCD complex of E. amylovora was compared with its counterpart from E. coli, using the FhuDBCD complex (7lb8) reported in the referenced research article. The superimposed structure of the complex has been added in Supplementary information.

Reviewer #3: Bharti et al. present a manuscript that evaluates predicted protein structure for an E. amylovora siderophore binding protein FhuD and binding analysis to a set of known siderophores. The manuscript uses alpha fold for structural predictions and then uses docking through autodock to compare FhuD to 6 siderophores. While the manuscript does a thorough analysis of the predicted structure of FhuD and the phylogenetic differences across various homologs, the docking predictions to the set of siderophores is not validated by binding analysis in vitro. Further, there is also a lack of the evaluation of a larger set of proteins from the E. amylovora genome for binding potential to the set of siderophores tested. These sets of information will be necessary to validate the docking predictions.

Comments are specifically provided below.

Table 3/Figure5: While these are predicted binding for the different ligands to FhuD, it needs to be validated by the recombinant expression of the FhuD and binding assays. Currently these are just predictions.

R: The inability to obtain a soluble recombinant protein posed a significant challenge, preventing experimental validation of the predicted interactions. Therefore, the presented data are solely based on in silico analyses. Molecular docking analyses were performed primarily as an indirect strategy to evaluate whether conserved residues, occupying equivalent structural positions across FhuD homologs, could mediate interactions with siderophores potentially utilized by Erwinia amylovora. This approach aimed to assess the plausibility of ligand recognition by FhuD_Ea based on structural alignment and the predicted spatial arrangement of key binding residues.

Figure 2/ Figure 5: Since the manuscript does not include data about the validated binding of FhuD to these ligands, another exercise would be to selectively modify the protein sequence to alter the potential binding pockets and rerun this through alpha fold and Autodock. To check if this can alter the binding as per the docking analysis.

R: While experimental validation of FhuD_Ea–ligand binding is beyond the scope of the current study, we performed in silico mutagenesis of key binding residues - specifically substituting them with alanine - to evaluate their contribution to ligand recognition. Molecular docking results revealed a notable decrease in binding affinity, particularly for Ferrioxamine, following these substitutions. This reduction is most likely due to the loss of critical interactions, such as hydrogen bonds and electrostatic contacts, that alanine cannot replace due to its minimal and non-polar side chain. Interestingly, in several cases, the docking simulations showed that, upon mutation, the ligands shifted slightly to form alternative interactions with deeper, more internal residues within the binding pocket. However, these compensatory interactions were weaker and less optimal, reinforcing the idea that the original, surface-exposed residues identified in our initial analysis play a fundamental and specific role in stabilizing the ligand.

Page 15 and Page 19 both have a Table 3. This might be a typo that needs to be modified.

R: This was indeed a typographical oversight. The table on page 19 has been renumbered appropriately to avoid duplication, and all in-text references have been updated accordingly.

Figure 4: While FhuD is the main transporter of interest in this study, due to the restricted nature of docking to a set group of siderophores, the data is mainly predictive and needs to be validated by binding studies. However, this data could be also supplemented by reverse docking analysis where the alpha fold predictions of the E. amylovora CFBP1430 genomic sequence can then be the protein database that each of the ligands get reverse docked into. This helps provide an unbiased outlook of the best candidates out of the E. amylovora genome that can bind to these siderophores.

R: While FhuD is the primary focus of this study, we acknowledge that the docking analysis is limited to a predefined set of siderophores and remains predictive in nature. Experimental validation, such as binding assays, will be essential to confirm these interactions. To further support the computational predictions, we agree that complementary strategies like reverse docking can provide valuable insights.

To explore this, we conducted a reverse docking experiment using the AlphaFold-predicted structure of EAMY_2771, a putative periplasmic binding protein encoded adjacent to fhuD_Ea. This approach allowed us to assess the interaction potential of siderophores with an alternative candidate from the E. amylovora CFBP1430 genome. The results suggest that while FhuD_Ea remains a strong candidate, other proteins may also exhibit moderate affinity, highlighting the usefulness of a further genome-wide reverse docking to identify additional siderophore-binding proteins.

Table 4: due to just the evaluation of FhuD docked agains

---

## [Decision Letter · Decision Letter 1]

Bioinformatics and modelling studies of FhuD, the periplasmic siderophore binding protein from the plant pathogen Erwinia amylovora

PONE-D-25-04770R1

Dear Dr. Benini,

We’re pleased to inform you that your manuscript has been judged scientifically suitable for publication and will be formally accepted for publication once it meets all outstanding technical requirements.

Kind regards,

Abhijeet Shankar Kashyap

Academic Editor

PLOS ONE

Additional Editor Comments (optional):

Reviewers' comments:

Reviewer's Responses to Questions

**Comments to the Author**

1. If the authors have adequately addressed your comments raised in a previous round of review and you feel that this manuscript is now acceptable for publication, you may indicate that here to bypass the “Comments to the Author” section, enter your conflict of interest statement in the “Confidential to Editor” section, and submit your "Accept" recommendation.

Reviewer #1: All comments have been addressed

Reviewer #2: All comments have been addressed

2. Is the manuscript technically sound, and do the data support the conclusions?

Reviewer #1: Yes

Reviewer #2: Yes

3. Has the statistical analysis been performed appropriately and rigorously? 

Reviewer #1: Yes

Reviewer #2: N/A

4. Have the authors made all data underlying the findings in their manuscript fully available?

Reviewer #1: Yes

Reviewer #2: Yes

5. Is the manuscript presented in an intelligible fashion and written in standard English?

Reviewer #1: Yes

Reviewer #2: Yes

6. Review Comments to the Author

Reviewer #1: The authors have addressed all my concerns and suggestions in this R1 manuscript.

Two minor corrections are needed:

Line 75: Use comma before "which" or use "that" instead of "which".

Table 2: "Seq%" (the far0right column): concert all values to %, e.g., 0.5373 to 53.7 and 0.188 to 18.8, and use one decimal point across the board.

Reviewer #2: The manuscript has been significantly improved, the authors have addressed all reviewer's comments, made the text of the paper clearer, and improved the content of the tables and the figures

7. PLOS authors have the option to publish the peer review history of their article (what does this mean? ). If published, this will include your full peer review and any attached files.

**Do you want your identity to be public for this peer review?** For information about this choice, including consent withdrawal, please see our Privacy Policy .

Reviewer #1: No

Reviewer #2: No

---

## [Editor Report · Acceptance letter]

PONE-D-25-04770R1

PLOS ONE

Dear Dr. Benini,

I'm pleased to inform you that your manuscript has been deemed suitable for publication in PLOS ONE. Congratulations! Your manuscript is now being handed over to our production team.

Kind regards,

on behalf of

Dr. Abhijeet Shankar Kashyap

Academic Editor

PLOS ONE